# Design, Modeling, and Testing of a Long-Stroke Fast Tool Servo Based on Corrugated Flexure Units

**DOI:** 10.3390/mi15081039

**Published:** 2024-08-15

**Authors:** Ning Chen, Zhichao Wen, Jiateng Rong, Chuan Tian, Xianfu Liu

**Affiliations:** 1School of Mechanical and Electronic Engineering, Shandong University of Science and Technology, Qingdao 266590, China; chenning3456@163.com (N.C.); wen_zhichao2000@163.com (Z.W.); rjiateng@163.com (J.R.); 2School of Mechanical Engineering, Shandong University, Jinan 250061, China; 3School of Automation Science and Electrical Engineering, Beihang University, Beijing 100191, China; 4School of Mechanical Engineering, Shandong University of Technology, Zibo 255000, China; lxfu2015@163.com

**Keywords:** compliant mechanism, corrugated flexure units, fast tool servo, micromachining, microstructure

## Abstract

To further enhance the performance of the fast tool servo (FTS) system in terms of stroke, load capacity, and application area, this paper proposes a novel fast tool servo device driven by a voice coil motor (VCM), based on a three-segment uniform corrugated flexure (CF) guiding mechanism, with a large stroke, high accuracy, and high dynamics. To describe the unified static characteristics of such device, the compliance matrix method is applied to establish its model, where the influence of CF beam structural parameters on the FTS device is investigated in detail. Furthermore, resolution and positioning accuracy tests are conducted to validate the features of the system. The testing results indicate that the maximum stroke of the FTS device is up to 3.5 mm and the positioning resolution values are 3.6 μm and 2.4 μm for positive and negative stroke, respectively, which further verifies the device’s effectiveness and promising application prospect in ultra-precision microstructure machining.

## 1. Introduction

With the continuous advancement of precision, complexity, and high efficiency in optical imaging, precision machining, and high-end equipment manufacturing areas, this has put forward higher requirements for micro and nanomachining manufacturing technologies [1,2] applied in optical lens surfaces [3,4], asymmetric surfaces [5], and micro/nano complex structures [6,7]. In particular, fast tool servo (FTS) is widely recognized as an effective method for ultra/high-precision machining due to its advantages such as rapid response speed, high processing efficiency, and excellent surface roughness [8], where precision driving methods encompass piezoelectric driving [9], magnetostrictive force driving [10], Maxwell force driving [11], voice coil motor (VCM) driving [12], and so on.

The piezoelectric-driven FTS system offers high precision and a fast response [13,14,15]. However, the limited output of its drive components restricts the stroke to typically tens to one or two hundred microns [16,17,18], posing challenges in processing the micro-structure of large-sized and large-curvature parts, for example, engine elliptical piston skirts with non-circular surfaces and deep groove optics that are not axisymmetric. Conversely, the FTS system driven by VCMs features a longer processing stroke [19,20].

Common FTS systems driven by VCMs usually use linear guides or flexible hinges as guiding mechanisms. For instance, the long-stroke FTS system using rolling guides and VCMs can reach a maximum stroke of ±4 mm [21]. The long-stroke FTS system with air bearing guiding and VCM driving achieves a 1% tracking error and a ±1 mm stroke [22,23,24]. The application of the VCM and linear guide rail can even increase the maximum stroke to 30 mm [25,26]. It should be pointed out that although the FTS systems actuated by VCMs, rolling guide rail, or air bearing can achieve a large stroke, there are large friction disturbances and guiding mechanism wear in the process of high-frequency machining, which deteriorates both the machining accuracy and the use life.

Obviously, a reasonable flexible mechanism design can not only avoid friction and wear, but also ensure that the micro-motion platform achieves very high motion accuracy and bandwidth [27,28]. The long-stroke FTS system with four flexible leaf springs has a maximum stroke of 2 mm [29], and the trajectory tracking error has reached 0.15% [30]. However, flexure hinges are typically arranged in a single plane, leading to system vibrations and compromised machining accuracy when subjected to significant vertical loads due to inadequate stiffness.

Similar to FTS systems, many precision micro-positioning platforms use a VCM-driven and flexible mechanism-guided design. For example, a two-degree of freedom flexible micro-positioning platform based on VCMs has a working space of 1.4 mm × 1.2 mm [31]. A flexible micro-clamp driven by a VCM, with an 8 mm grasp range and a 1 μm displacement resolution, accurately achieves the operation task [32]. Notably, the micro-positioning platform, utilizing the VCMs and the CF beam, exhibits a significantly expanded operational range (2 mm/28 N) compared to conventional micro-positioning platforms constructed with ordinary flexible hinges [33,34].

In summary, a long-stroke FTS system using a VCM and a flexible guiding mechanism is ideal for machining complex parts, such as deep groove optics and elliptical piston skirts for engines, and different flexible hinge types and arrangement structures can significantly affect the working performance of the guiding mechanism. However, the existing flexible mechanisms and hinge arrangement structures do not work well as guiding mechanisms for long-stroke FTS systems. The common flexible lever mechanism can amplify the driving stroke, but it generates motion hysteresis and tracking errors; the conventional straight beam flexible mechanism and the right circular hinge flexible mechanism have relatively small motion strokes. The CF mechanism, on the other hand, not only has a longer traveling stroke, but also has the potential to vary the stiffness over a wide range in order to modulate the intrinsic frequency of the system. In addition, the three-section uniformly arranged flexible beams have a higher guiding stiffness compared to the conventional planar arrangement of flexible beams.

In order to further enhance the stroke capability and load-carrying capacity of the flexible guiding mechanism of the FTS system, this paper adopts a VCM as the actuator and innovatively utilizes an assembly-type three-section uniform CF guide plate to construct a long-stroke and high-stiffness FTS system. To describe the unified static characteristics of such device, the compliance matrix method is applied to establish its model, and the influence of CF beam structural parameters on the FTS device is investigated. Moreover, an experimental platform based on the five-link long-stroke precision FTS system is built, enabling a series of tests for function and performance verification. Compared with the traditional FTS system, it has a longer machining stroke and motion stability, which provides novel insights into designing flexible guiding mechanisms for long-stroke FTS systems.

## 2. Design of FTS Platform with CF Guiding Mechanism

As shown in Figure 1, the novel FTS device presented in this paper mainly consists of a tool, a three-segment uniformly distributed CF guiding mechanism (three CF beams distributed at an angle of 120°), a voice coil motor, and a grating sensor, where the performance of the device (e.g., stroke and intrinsic frequency) can be user-defined by replacing the appropriate flexible guide plate with different structural parameters, allowing for adjustments in the working stroke and natural frequency to meet various processing requirements in different scenarios. Moreover, the CF guiding mechanism is actuated by the VCM to achieve a reciprocating motion, with the grating detector serving as the displacement sensor to detect the displacement of the tool in real time.

## 3. Static Modeling of CF Guiding Mechanism

The present study focuses on the modeling approach for a CF guiding mechanism with multiple semi-circle flexure units. Specifically, the compliance matrix method is employed to perform static modeling of the CF guiding mechanism, enabling the solution of mathematical expressions that describe the relationship between driving force and motion displacement in the FTS system. Moreover, this paper discusses in detail the derivation process of the statics model from semi-circle flexure units to encompassing the entire CF guiding mechanism and presents a general mechanical model applicable to CF beams with any number of links.

### 3.1. Flexibility Model for Semi-Circle Flexure Units

The force in the working process of a CF beam is entirely in the plane force, thus necessitating a planar approach for modeling the flexibility of each component of the CF beam. The Mohr’s Integral Method is employed in this paper to derive the compliance matrix of the semi-circle flexure element in the plane, as shown in Equation (1), which is used to calculate the vertical displacement *δ*_x_, horizontal displacement *δ*_y,_ and rotation angle *θ*_z_ at the free end of the semi-circle flexure unit, where one end of the semi-circle flexure unit is fixed and the other end is applied with the conditions of vertical force *F*_x_, horizontal force *F*_y,_ and bending moment *M*_z_, respectively, as shown in Figure 2. Since the CF beam mainly undergoes bending deformation after being stressed, the effects of axial force and torque on the displacement of the free end can be neglected when applying the Mohr’s Integral Method for calculation.

Next, the radius of the semi-circle flexure unit is set to be r, and the polar angle is *θ*. The distance between any point on the semi-circle flexure unit and the origin of the X_2_, Y_2_ coordinate system can be calculated by *x* = *r*(1 + cos*θ*), *y* = *r*sin*θ*. Taking the force in Figure 2a as an example, the bending moment at any position is *M*(*θ*), and then, assuming that the unit force vertically downward and horizontally rightward and the unit bending moment counterclockwise are applied to the free end, the bending moments of the semi-circle flexure unit under the action of different units of force are expressed as $\bar{M}(\theta {)}_{x}$, $\bar{M}(\theta {)}_{y}$, and $\bar{M}(\theta {)}_{m}$, respectively, as shown in Equation (2).
(1)$$\Delta =\int_{l}\frac{\bar{M}(x)\cdot M(x)}{EI}dx$$
(2)$$\left\{\begin{array}{l} M(\theta )=r(1+\cos\theta )F_{x}\\ \bar{M}(\theta {)}_{x}=r(1+\cos\theta )\\ \bar{M}(\theta {)}_{y}=-r\sin\theta \\ \bar{M}(\theta {)}_{m}=-1 \end{array}\right.$$

The elastic modulus of the semi-circle flexure unit is denoted as *E*, and its moment of inertia is represented by *I*, where the moment of inertia can be calculated by *I* = (*b* × *h*^3^)/12, with *b* and *h* being the section width and thickness of the semi-circle flexure unit. The displacements *δ*_x1_, *δ*_y1_, and *θ*_1_ generated at the free end of the semi-circle flexure unit are calculated using the Mohr’s Integral Method, as shown in Equation (3), where *dl* can be expressed by Equation (4).
(3)$$\left\{\begin{array}{l} \delta_{x1}=\int_{0}^{\pi }\frac{\bar{M}(\theta {)}_{x}\cdot M(\theta )}{EI}dl=\frac{3\pi r^{3}}{2EI}F_{x}\\ \delta_{y1}=\int_{0}^{\pi }\frac{\bar{M}(\theta {)}_{y}\cdot M(\theta )}{EI}dl=-\frac{2r^{3}}{EI}F_{x}\\ \theta_{1}=\int_{0}^{\pi }\frac{\bar{M}(\theta {)}_{m}\cdot M(\theta )}{EI}dl=-\frac{\pi r^{2}}{EI}F_{x} \end{array}\right.$$
(4)$$dl=\sqrt{{\left(\frac{dx}{d\theta }\right)}^{2}+{\left(\frac{dy}{d\theta }\right)}^{2}}d\theta =rd\theta$$

Similarly, the displacements *δ*_x2_, *δ*_y2_, and *θ*_2_ and *δ*_x3_, *δ*_y3_, and *θ*_3_ at the free end of the semi-circle flexure unit can be calculated under the conditions of the horizontal force *F*_y_ and the bending moment *M*_z_ applied at the free end of the semi-circle flexure unit.

The relationship between displacement Δ, flexibility *C*, and force *F* can be obtained from the equation Δ=C·F, and then, the compliance matrix *C*_12_ of the semi-circle flexure unit can be derived, as shown in Equation (5).
(5)$$C_{12}=\left(\begin{array}{ccc} \frac{3\pi r^{3}}{2EI} & -\frac{2r^{3}}{EI} & -\frac{\pi r^{2}}{EI}\\ -\frac{2r^{3}}{EI} & \frac{\pi r^{3}}{2EI} & \frac{2r^{2}}{EI}\\ -\frac{\pi r^{2}}{EI} & \frac{2r^{2}}{EI} & \frac{\pi r}{EI} \end{array}\right)$$

### 3.2. Flexibility Modeling of CF Units

The CF unit can be considered as comprising the semi-circle flexure unit O_1_O_2_ and the semi-circle flexure unit O_2_O_3_, as illustrated in Figure 3, where the semi-circle flexure unit O_2_O_3_ can be obtained by rotating O_1_O_2_ around the node O_2_ by 180°. Therefore, the semi-circle flexure units O_1_O_2_ and O_2_O_3_ have an equal compliance matrix in their respective coordinate systems O_2_X_2_Y_2_ and O′_2_X′_2_Y′_2_, that is, *C*_12_ = *C*_23_. After this, the compliance matrix *C*′_12_ and *C*′_23_ of the semi-circle flexure units O_1_O_2_ and O_2_O_3_ in the O_3_X_3_Y_3_ coordinate system can be obtained by the corresponding coordinate transformations (translations and rotations) as follows:(6)$${C^{\prime }}_{12}={T_{12}}^{T}\cdot {R_{12}}^{T}\cdot C_{12}\cdot R_{12}\cdot T_{12},$$
(7)$${C^{\prime }}_{23}={T_{23}}^{T}\cdot {R_{23}}^{T}\cdot C_{23}\cdot R_{23}\cdot T_{23}.$$

Among them, the translation matrix *T*_12_ and rotation matrix *R*_12_ of the semi-circle flexure unit O_1_O_2_ are shown in Equations (8) and (9), respectively; the translation matrix *T*_23_ = *T*_12_ and rotation matrix *R*_23_ of the semi-circle flexure unit O_2_O_3_ are shown in Equation (10).
(8)$$T_{12}=\left(\begin{array}{ccc} 1 & 0 & 0\\ 0 & 1 & 0\\ \Delta_{y} & -\Delta_{x} & 1 \end{array}\right)=\left(\begin{array}{ccc} 1 & 0 & 0\\ 0 & 1 & 0\\ 2r & 0 & 1 \end{array}\right)$$
(9)$$R_{12}=\left(\begin{array}{ccc} \cos\theta & \sin\theta & 0\\ -\sin\theta & \cos\theta & 0\\ 0 & 0 & 1 \end{array}\right)=\left(\begin{array}{ccc} -1 & 0 & 0\\ 0 & -1 & 0\\ 0 & 0 & 1 \end{array}\right)$$
(10)$$R_{23}=\left(\begin{array}{ccc} 1 & 0 & 0\\ 0 & 1 & 0\\ 0 & 0 & 1 \end{array}\right)$$

Subsequently, the compliance matrix of the two semi-circle flexure units in the O_3_X_3_Y_3_ coordinate system can be superimposed to obtain the compliance matrix *C*_123_ of the CF unit O_1_O_2_O_3_, as follows:(11)$$C_{123}={C^{\prime }}_{12}+{C^{\prime }}_{23}=\left(\begin{array}{ccc} \frac{11\pi r^{3}}{EI} & -\frac{4r^{3}}{EI} & \frac{4\pi r^{2}}{EI}\\ -\frac{4r^{3}}{EI} & \frac{\pi r^{3}}{EI} & 0\\ \frac{4\pi r^{2}}{EI} & 0 & \frac{2\pi r}{EI} \end{array}\right)$$

### 3.3. Flexibility Modeling of Five-Link CF Beams

The compliance matrix of the five-link CF beam is derived. As depicted in Figure 4, the CF beam consists of the CF units O_1_O_2_O_3_ and O_3_O_4_O_5_ and the semi-circle flexure unit O_5_O_6_ in series, and the compliance matrices of the CF units O_1_O_2_O_3_ and O_3_O_4_O_5_ are equivalent within their respective coordinate systems O_3_X_3_Y_3_ and O_5_X_5_Y_5_. Similarly, the compliance matrices for the semi-circle flexure units O_5_O_6_ and O_1_O_2_ are equal within their respective coordinate systems O_2_X_2_Y_2_ and O_6_X_6_Y_6_, namely, *C*_345_ = *C*_123_, *C*_56_ = *C*_12_. Consequently, by converting the compliance matrix from each coordinate system on the plane to the corresponding coordinates in the system (O_6_X_6_Y_6_), we can obtain the compliance matrix for each unit.

In particular, the rotation matrix *R*_36_ = *R*_12_ and translation matrix *T*_36_ of the CF unit O_1_O_2_O_3_ are shown in Equation (12); the rotation matrix *R*_56_ = *R*_12_ and translation matrix *T*_56_ of the CF unit O_3_O_4_O_5_ are shown in Equation (13); the semi-circle flexure unit O_5_O_6_ does not necessitate coordinate transformation.
(12)$$T_{36}=\left(\begin{array}{ccc} 1 & 0 & 0\\ 0 & 1 & 0\\ -6r & 0 & 1 \end{array}\right)$$
(13)$$T_{56}=\left(\begin{array}{ccc} 1 & 0 & 0\\ 0 & 1 & 0\\ -2r & 0 & 1 \end{array}\right)$$

The final compliance matrix *C*_16_ for the CF beam is derived as follows:(14)$$\begin{array}{ll} C_{16} & ={T_{36}}^{T}\cdot {R_{36}}^{T}\cdot C_{123}\cdot R_{36}\cdot T_{36}+{T_{56}}^{T}\cdot {R_{56}}^{T}\cdot C_{345}\cdot R_{56}\cdot T_{56}+C_{56}\\ & =\left(\begin{array}{ccc} \frac{335\pi r^{3}}{2EI} & -\frac{10r^{3}}{EI} & -\frac{25\pi r^{2}}{EI}\\ -\frac{10r^{3}}{EI} & \frac{5\pi r^{3}}{2EI} & \frac{2r^{2}}{EI}\\ -\frac{25\pi r^{2}}{EI} & \frac{2r^{2}}{EI} & \frac{5\pi r}{EI} \end{array}\right) \end{array}$$

### 3.4. Static Modeling of the FTS System

In order to derive the compliance matrix of the CF guiding mechanism, the displacement of the guide mechanism is influenced by three pairs of mirror-image-arranged CF beams, which can be categorized into three groups (I, II, and III) for research purposes, as illustrated in Figure 5. The compliance matrices for each set of coordinate systems (OXYZ, OX′Y′Z′, OX″Y″Z″) are denoted by *C*_1_, *C*_2_, and *C*_3_, respectively. It should be noted that all three coordinate systems share an X-axis.

In the following, the compliance matrix of a pair of CF beams in the overall coordinate system OXY of Figure 6 (in which the Z-axis is perpendicular to the paper surface outward) is investigated by taking the CF beams in the mirror arrangement of Group I as an example. It can be observed that the compliance matrix *C*_16_ of the CF beam O_1_O_6_ in the coordinate system O_6_X_6_Y_6_ is equal to the compliance matrix *C*′_16_ of the CF beam O′_1_O′_6_ in the coordinate system O′_6_X′_6_Y′_6_, that is, *C*_16_ = *C*′_16_.

Then, the compliance matrix of the CF beams O_1_O_6_ and O′_1_O′_6_ under the overall coordinate system OXY can be obtained by coordinate transformation. It is worth noting that there is a parallel structure between O_1_O_6_ and O′_1_O′_6_; therefore, the stiffness model of a set of CF beams under the overall coordinate system OXY can be expressed as follows:(15)$$K_{1}=({T_{1}}^{T}\cdot {R_{1}}^{T}\cdot C_{16}\cdot R_{1}\cdot T_{1}{)}^{-1}+({T_{2}}^{T}\cdot {R_{2}}^{T}\cdot {C^{\prime }}_{16}\cdot R_{2}\cdot T_{2}{)}^{-1}$$

Moreover, the rotation matrix *R*_1_ of the CF beam O_1_O_6_ is a unit matrix, and the translation matrix *T*_1_ is shown in Equation (16); for the rotation matrix *R*_2_ = *R*_12_ of the CF beam O′_1_O′_6_, the translation matrix *T*_2_ is shown in Equation (17).
(16)$$T_{1}=\left(\begin{array}{ccc} 1 & 0 & 0\\ 0 & 1 & 0\\ -\Delta_{y} & \Delta_{x2} & 1 \end{array}\right)$$
(17)$$T_{2}=\left(\begin{array}{ccc} 1 & 0 & 0\\ 0 & 1 & 0\\ -(\Delta_{y}+10r) & \Delta_{x1}+\Delta_{x2} & 1 \end{array}\right)$$

The compliance matrix (*C*_1_) and stiffness matrix (*K*_1_) of Group I are inversely proportional to each other, and the compliance matrix of the three groups of CF beams is the same in their respective coordinate systems, namely, *C*_1_ = *C*_2_ = *C*_3_.

Hence, the compliance matrices of the three groups in their respective coordinate systems are unified into the overall coordinate system OXYZ, where the rotation matrix *R*_II_ of the coordinate systems OX′Y′Z′ to OXYZ can be expressed by Equation (18), the rotation matrix *R*_III_ of the coordinate systems OX″Y″Z″ to OXYZ can be expressed by Equation (19), and the overall stiffness *K* of the CF guiding mechanism can be calculated by Equation (20).
(18)$$R_{\mathrm{II}}=\left(\begin{array}{cccccc} 1 & 0 & 0 & 0 & 0 & 0\\ 0 & -\frac{1}{2} & -\frac{\sqrt{3}}{2} & 0 & 0 & 0\\ 0 & \frac{\sqrt{3}}{2} & -\frac{1}{2} & 0 & 0 & 0\\ 0 & 0 & 0 & 1 & 0 & 0\\ 0 & 0 & 0 & 0 & -\frac{1}{2} & -\frac{\sqrt{3}}{2}\\ 0 & 0 & 0 & 0 & \frac{\sqrt{3}}{2} & -\frac{1}{2} \end{array}\right)$$
(19)$$R_{\mathrm{III}}=\left(\begin{array}{cccccc} 1 & 0 & 0 & 0 & 0 & 0\\ 0 & -\frac{1}{2} & \frac{\sqrt{3}}{2} & 0 & 0 & 0\\ 0 & -\frac{\sqrt{3}}{2} & -\frac{1}{2} & 0 & 0 & 0\\ 0 & 0 & 0 & 1 & 0 & 0\\ 0 & 0 & 0 & 0 & -\frac{1}{2} & \frac{\sqrt{3}}{2}\\ 0 & 0 & 0 & 0 & -\frac{\sqrt{3}}{2} & -\frac{1}{2} \end{array}\right)$$
(20)$$K={C_{1}}^{-1}+{\left({R_{\mathrm{II}}}^{T}\cdot C_{2}\cdot R_{\mathrm{II}}\right)}^{-1}+{\left({R_{\mathrm{III}}}^{T}\cdot C_{3}\cdot R_{\mathrm{III}}\right)}^{-1}$$

Finally, the static model of the FTS system based on the compliance matrix method is established as follows:(21)δ=C·F
where *δ* represents the stroke of the FTS system.

### 3.5. Static Modeling of FTS Systems with Arbitrary Number of Links

The purpose of this section is to derive the general statics model of an FTS system with any number of links in order to further investigate its displacement range. This derivation assumes that the total length of the CF beam, denoted as L, remains constant.

(1) When the number of links *n* is odd, as shown in Figure 7, the CF beam can be regarded as composed of (*n* − 1)/2 CF units and a semi-circle flexure unit in series, and the radius *r* of the links can be expressed by L/(2*n*) at this time. Therefore, the compliance matrix in each local coordinate system can be converted to the overall coordinate system (O_n+1_X_n+1_Y_n+1_) at the end of the CF beam and superimposed. At this time, the translation matrix *T*_2k+1 n+1_ of the *k*th CF unit is shown in Equation (22). Meanwhile, we can derive the compliance matrix *C*_n n+1_ = *C*_12_, and *C*_n−2 n−1 n_ = …… = *C*_345_ = *C*_123_. Hence, the compliance matrix *C*_1 n+1_ of CF beams under an odd number of links can be obtained by superposition of the compliance matrix, as shown in Equation (23).
(22)$$T_{2k+1\;n+1}=\begin{array}{cc} \left(\begin{array}{ccc} 1 & 0 & 0\\ 0 & 1 & 0\\ -\left\{\left[\frac{\left(n-1\right)}{2}-k\right]\cdot 4r+2r\right\} & 0 & 1 \end{array}\right) & \begin{array}{l} n=1,3,5,7\ldots \ldots \\ k\leq \frac{n-1}{2} \end{array} \end{array}$$
(23)$$\begin{array}{l} C_{1\;n+1}={T_{3\;n+1}}^{T}\cdot {R_{3\;n+1}}^{T}\cdot C_{123}\cdot R_{3\;n+1}\cdot T_{3\;n+1}+{T_{5\;n+1}}^{T}\cdot {R_{5\;n+1}}^{T}\cdot C_{345}\cdot R_{5\;n+1}\cdot T_{5\;n+1}+\ldots \\ +{T_{n\;n+1}}^{T}\cdot {R_{n\;n+1}}^{T}\cdot C_{n-2\;n-1\;n}\cdot R_{n\;n+1}\cdot T_{n\;n+1}+C_{n\;n+1}\\ =\left(\begin{array}{ccc} \frac{\pi r^{3}}{EI}\left(\frac{4}{3}n^{3}+\frac{1}{6}n\right) & -\frac{2nr^{3}}{EI} & -\frac{n^{2}\pi r^{2}}{EI}\\ -\frac{2nr^{3}}{EI} & \frac{n\pi r^{3}}{2EI} & \frac{2r^{2}}{EI}\\ -\frac{n^{2}\pi r^{2}}{EI} & \frac{2r^{2}}{EI} & \frac{n\pi r}{EI} \end{array}\right)\begin{array}{c} \;n=1,3,5,7\ldots \ldots \end{array} \end{array}$$

(2) When the number of links *n* is even, as shown in Figure 8, the CF beam can be regarded as consisting of *n*/2 CF units connected in series, with the radius *r* represented in the same way as the odd-numbered links, and *C*_n−1 n n+1_ = …… = *C*_345_ = *C*_123_ in the respective coordinate system. Similarly, the translation matrix *T*_2k+1 n+1_ of the *k*th CF unit is shown in Equation (24), and the compliance matrix of the CF beam with an even number of links can be expressed by Equation (25).
(24)$$T_{2k+1\;n+1}=\begin{array}{cc} \left(\begin{array}{ccc} 1 & 0 & 0\\ 0 & 1 & 0\\ \left(\frac{n}{2}-k\right)\cdot 4r & 0 & 1 \end{array}\right) & \begin{array}{l} n=2,4,6,8\ldots \ldots \\ k\leq \frac{n}{2} \end{array} \end{array}$$
(25)$$\begin{array}{l} C_{1\;n+1}={T_{3\;n+1}}^{T}\cdot {R_{3\;n+1}}^{T}\cdot C_{123}\cdot R_{3\;n+1}\cdot T_{3\;n+1}+{T_{5\;n+1}}^{T}\cdot {R_{5\;n+1}}^{T}\cdot C_{345}\cdot R_{5\;n+1}\cdot T_{5\;n+1}+\ldots \\ +{T_{n+1\;n+1}}^{T}\cdot {R_{n+1\;n+1}}^{T}\cdot C_{n-1\;n\;n+1}\cdot R_{n+1\;n+1}\cdot T_{n+1\;n+1}\\ =\left(\begin{array}{ccc} \frac{\pi r^{3}}{EI}\left(\frac{4}{3}n^{3}+\frac{1}{6}n\right) & -\frac{2nr^{3}}{EI} & \frac{n^{2}\pi r^{2}}{EI}\\ -\frac{2nr^{3}}{EI} & \frac{n\pi r^{3}}{2EI} & 0\\ \frac{n^{2}\pi r^{2}}{EI} & 0 & \frac{n\pi r}{EI} \end{array}\right)\begin{array}{c} \;\;n=2,4,6,8\ldots \ldots \end{array} \end{array}$$

Finally, the compliance matrix *C*′_1_ in the OXY coordinate system can be obtained by parallel superposition of a group of CF beams that are mirrored in the plane, as depicted in Equation (26), where *C*′_1 n+1_ = *C*_1 n+1_, *T*_3_ is shown in Equation (27), *T*_4_ is shown in Equation (28), *R*_3_ = *R*_12,_ and *R*_4_ is shown in Equation (29). Subsequently, the compliance matrix of the three groups of CF beams is consolidated into the global coordinate system OXYZ, yielding the compliance matrix *C*′ for any link within the CF guiding mechanism, as shown in Equation (30), where *C*′_1_ = *C*′_2_ = *C*′_3_.
(26)$${C^{\prime }}_{1}=\left\{\begin{array}{c} {\left[({T_{1}}^{T}\cdot {R_{1}}^{T}\cdot C_{1\;n+1}\cdot R_{1}\cdot T_{1}{)}^{-1}+({T_{3}}^{T}\cdot {R_{2}}^{T}\cdot {C^{\prime }}_{1\;n+1}\cdot R_{2}\cdot T_{3}{)}^{-1}\right]}^{-1}\;\;n=1,3,5,7\ldots \ldots \\ {\left[({T_{1}}^{T}\cdot {R_{3}}^{T}\cdot C_{1\;n+1}\cdot R_{3}\cdot T_{1}{)}^{-1}+({T_{4}}^{T}\cdot {R_{4}}^{T}\cdot {C^{\prime }}_{1\;n+1}\cdot R_{4}\cdot T_{4}{)}^{-1}\right]}^{-1}\;\;n=2,4,6,8\ldots \ldots \end{array}\right.$$
(27)$$T_{3}=\left(\begin{array}{ccc} 1 & 0 & 0\\ 0 & 1 & 0\\ -(\Delta_{y}+2n\cdot r) & \Delta_{x1}+\Delta_{x2} & 1 \end{array}\right)$$
(28)$$T_{4}=\left(\begin{array}{ccc} 1 & 0 & 0\\ 0 & 1 & 0\\ -\Delta_{y} & \Delta_{x1}+\Delta_{x2} & 1 \end{array}\right)$$
(29)$$R_{4}=\left(\begin{array}{ccc} 1 & 0 & 0\\ 0 & -1 & 0\\ 0 & 0 & -1 \end{array}\right)$$
(30)$$C^{\prime }={\left[{{C^{\prime }}_{1}}^{-1}+{\left({R_{\mathrm{II}}}^{T}\cdot {C^{\prime }}_{2}\cdot R_{\mathrm{II}}\right)}^{-1}+{\left({R_{\mathrm{III}}}^{T}\cdot {C^{\prime }}_{3}\cdot R_{\mathrm{III}}\right)}^{-1}\right]}^{-1}$$

In summary, the unified static model of the FTS system under arbitrary links can be expressed as follows:(31)$$\delta =C^{\prime }\cdot F$$

## 4. Static Model Validation and Static Characteristic Experiments

This study involved the construction of a five-link FTS system experimental platform based on the structural parameters provided in Table 1. The accuracy of the statics model was validated by finite element analysis (FEA) and stroke testing, encompassing positioning and repeated-positioning evaluations.

### 4.1. Verification of the Accuracy of the Static Model

In this section, the displacements of the end of the semi-circle flexure unit, CF unit, and CF beam under different forces were firstly simulated by using the software ABAQUS and compared with the theoretical values of the modeled ones in order to validate the correctness. Concretely, the stroke of the FTS platform based on a three-section homogeneous CF guiding mechanism was simulated under driving force (−200 N to 200 N). Finally, the accuracy of the static model of the long-stroke FTS system was verified through the comparison between the theoretical values, simulation results, and experimental data.

#### 4.1.1. FEA Verification

The material parameters of the CF guiding mechanism are presented in Table 2. By comparing the simulation values with the theoretical values, the semi-circle flexure unit has a relatively small error of approximately 3.5% in the x direction, and a larger error of around 10.4% in the y direction; the analytical solution of the CF unit is closer to the simulation value in the x direction, with an error of 3.7%, while the error in the y direction is 9.5%. Figure 9 shows the comparison between the theoretical and simulation results of the displacement of the CF beam in the x and y directions under the force *F*_x_, with the error of the displacement in the x direction not exceeding 4.7% and that in the y direction not exceeding 5.2%, which validates the derivation method proposed in this paper for constructing a static flexibility model.

Finally, the stroke of the FTS system under different driving forces can be obtained by simulation, and the displacement cloud obtained by simulation is shown in Figure 10, while the relationship between the simulation values and the theoretical values is shown in Figure 11. It can be seen that the relative error is below 2%, thereby providing preliminary validation for the statics model of the long-stroke FTS system based on the CF guiding mechanism.

#### 4.1.2. Stroke Test Experiments

In order to further validate the accuracy between the theoretical and actual displacements of the static model of the FTS platform, a long-stroke FTS system was built with a five-link flexible guiding mechanism, as depicted in Figure 12. The VCM was controlled by the output test signal from the computer to achieve reciprocating motion in both forward and backward directions, and the actual displacement output of the FTS platform was detected using a grating ruler with a resolution of 20 nm. The experimental principle is shown in Figure 13.

The experimental results are shown in Figure 11. The average error of the negative traveling direction is 5.4%, the average error of the positive traveling direction is 3.5%, where the maximum error of both directions is not more than 5.9%, and the error of the positive and negative directions gradually increases with the increase in the stroke. Notably, this paper demonstrates that our established statics model for the long-stroke FTS system exhibits proven accuracy and effectiveness through the comparison between the theoretical values, simulation results, and experimental data.

### 4.2. Experiments on Static Characteristics under Open-Loop Control

#### 4.2.1. System Resolution Test

In order to obtain the minimum resolution of positive and negative strokes, a step signal with increasing amplitude was input into the FTS system until both positive and negative strokes made a complete displacement response, as shown in Figure 14. The figure illustrates that the overall displacement of the experiment is minimal, with a discrepancy in resolution between the positive and negative directions. Upon increasing the amplitude of the displacement signal to 2.4 μm, a complete response is observed in the negative direction while lacking in the positive direction, as depicted in Figure 14a. Subsequently, by further increasing the displacement signal to 3.6 μm, both positive and negative directions exhibit full responsiveness to the displacement signal, as shown in Figure 14b.

#### 4.2.2. Positioning Accuracy and Repeat Positioning Accuracy Test

To evaluate the positioning accuracy and repeat positioning accuracy of the FTS system under open-loop control, experiments were conducted as depicted in Figure 15. The experimental data were processed to generate images showing the deviation of the actual displacement from the theoretical displacement within a 2 mm working stroke. The FTS system moves 1 mm from the 0 mm position in 0.2 mm steps and returns to 0 mm in positive and negative traveling directions; Figure 15a shows the positioning and repetitive positioning test for negative travel and Figure 15b shows the positioning and repetitive positioning test for positive travel.

By analyzing the data, it can be obtained that the positioning accuracy of the negative working stroke of the system is 12.8 μm/200 μm, and the repetitive positioning accuracy is 0.47 μm; the positioning accuracy of the positive working stroke is 7.9 μm/200 μm, and the repetitive positioning accuracy is 0.49 μm. Therefore, the positioning accuracy of the positive stroke is better than that of the negative stroke. In addition, the repeated positioning accuracy of the positive and negative strokes indicates that the CF guiding mechanism exhibits exceptional motion precision and ensures excellent workpiece consistency.

## 5. Discussion

This study explores the impact of structure parameters of the CF guiding mechanism on the platform stroke, which can be applied to the optimal design of the FTS system. In particular, this article investigates the influence of three structural parameters, namely width (*b*), thickness (*h*), and number of links (*n*), of the CF beam on the guide plate based on a static model of the CF guiding mechanism with any number of links. The stroke of the FTS system is examined under the condition that the total length (L) of the CF beam remains fixed and the driving force is 200 N. The results are illustrated in Figure 16.

The figure demonstrates the following laws:

(1) With a fixed number of links *n*, reducing the thickness *h* and width *b* of the flexible hinge leads to an increased stroke in the FTS system. This is due to the fact that the reduction in the thickness *h* and width *b* causes the moment of inertia *I* to decrease. From Equation (14), it can be seen that the reduction in the moment of inertia *I* leads to an increase in the compliance of the CF beam, which in turn leads to an increase in the traveling distance of the FTS system.

(2) By maintaining the thickness *h*, width *b*, and total length of the hinge, increasing the number of links *n* will reduce the stroke difference between the odd and even hinges until it reaches a stable value. This is owing to the fact that when the total length of the CF beam is kept constant, the radius of the semi-circle flexure unit gradually decreases as the number of links *n* increases, and when the radius decreases to a sufficiently small size, the CF beam can be approximated to a special straight beam, at which time the number of links *n* does not have much influence on the stroke.

(3) Amongst these three structural designs influencing the stroke, thickness *h* has the most significant impact on FTS system motion stroke. This is because, as seen from *I* = (*b* × *h*^3^)/12, the thickness *h* is a cubic term in the calculation and has a greater effect than the width *b*. In addition, the thickness *h* is of a different order of magnitude than the number of links *n* and has the most significant effect on the stroke. This can also be visualized in Figure 16.

(4) The odd number of links *n* in CF beams is more favorable for enhancing platform travel under the condition of a constant total length of the CF beam, which is verified by FEA.

Based on the relationships between different structures and the motion stroke in the FTS system, the structural design scheme with the best performance under different working conditions can be obtained, and the suitable CF guide plate can be replaced to meet processing requirements for different workpieces.

It is worth mentioning that CF guide plates with different configurations not only affect the motion travel of the FTS system, but also change the dynamic characteristics of the system, which is also crucial for the application of FTS in ultra-precision and micro-cutting. For the same driving force, a guide plate with a smaller stroke has a larger intrinsic frequency. The first-order intrinsic frequency of the long-stroke FTS system based on a five-link guide plate is experimentally tested to be 83 Hz.

However, although the current design can be used to adjust the intrinsic frequency of the FTS system by replacing the CF guiding plates with different structures, there are still limitations in the adjustment method: the adjustment process is cumbersome, and several sets of CF guiding plates with different structures need to be prepared to meet the replacement requirements.

## 6. Conclusions

This paper proposed a long-stroke FTS system which leveraged the large deformation, high precision, and frictionless characteristics of CF beams to design an innovative CF guiding mechanism featured with a high load capacity. The static model of the system was established using the compliance matrix method, and the validity of the model was confirmed by the test of a five-link FTS system experimental platform. The test results indicated that the maximum effective stroke was 3.5 mm, the resolution for positive and negative strokes was 3.6 μm and 2.4 μm, respectively, the overall positioning accuracy was 48.8 μm/200 μm, the repeated positioning accuracy was 1.6 μm, and the first-order intrinsic frequency was 83 Hz. This study presented a novel concept and an accurate statistical model for the design of the three-segment uniform CF guiding mechanism, thereby achieving exceptional precision and simultaneously increasing its stroke length. Future development plans involve the design of a long-stroke FTS system that utilizes the unique properties of the CF beam to enable easy stiffness adjustment, allowing for a wide range of variation in stiffness with an axial load.

## Figures and Tables

**Figure 1 micromachines-15-01039-f001:**
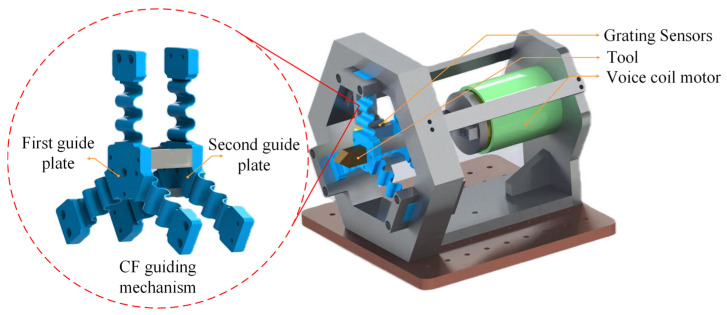
Long-stroke FTS device based on assembled three-segment homogeneous CF guide plate.

**Figure 2 micromachines-15-01039-f002:**
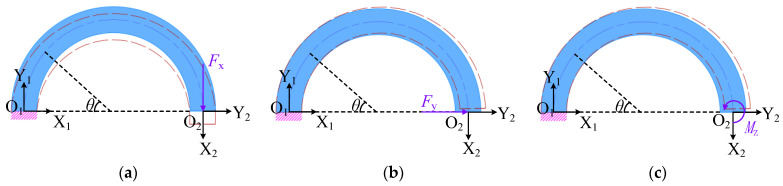
Semi-circle flexure unit under plane force: (**a**) external force *F*_x_; (**b**) external force *F*_y_; (**c**) bending moment *M*_z_.

**Figure 3 micromachines-15-01039-f003:**
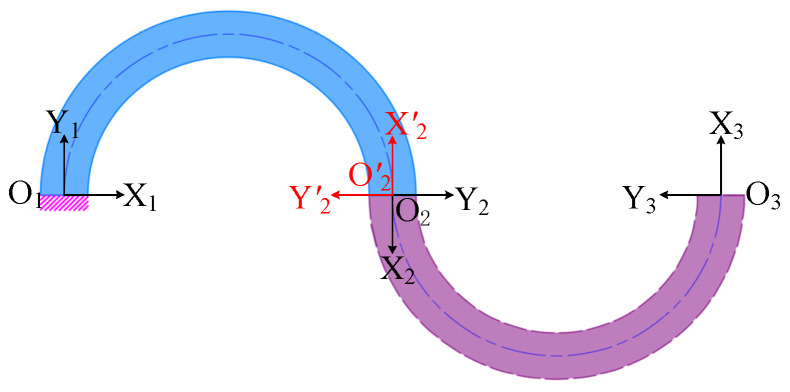
CF unit.

**Figure 4 micromachines-15-01039-f004:**
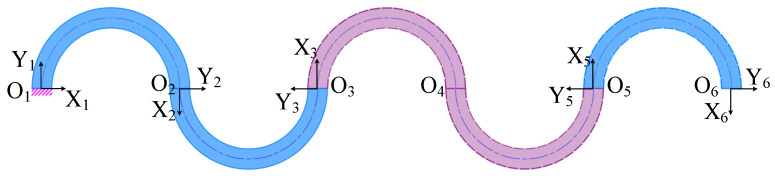
Five-link CF beams.

**Figure 5 micromachines-15-01039-f005:**
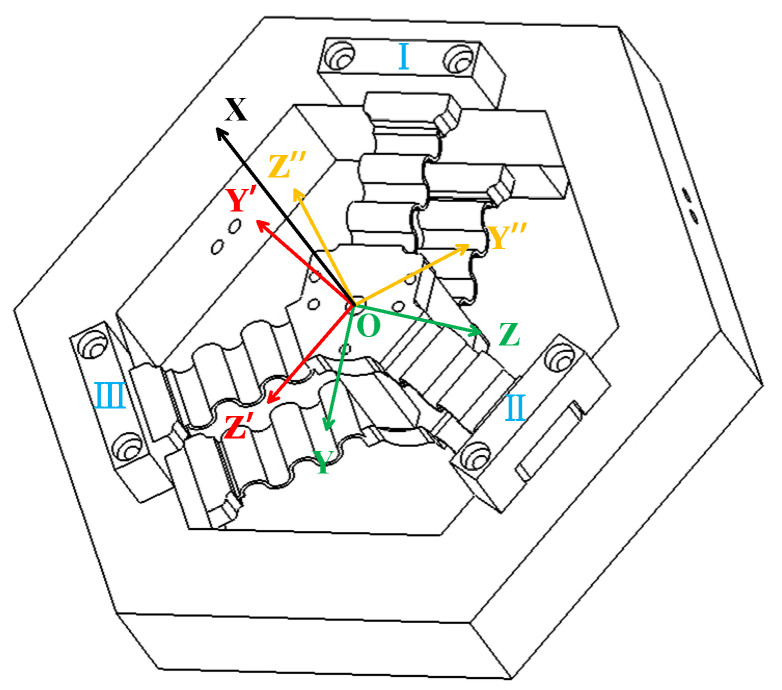
CF guiding mechanism.

**Figure 6 micromachines-15-01039-f006:**
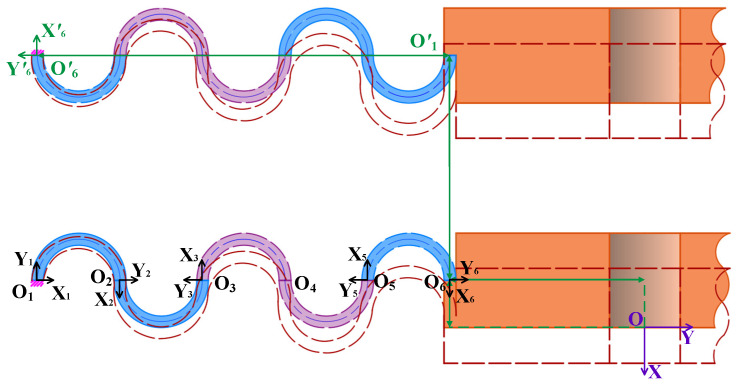
CF beams in mirror-image arrangement.

**Figure 7 micromachines-15-01039-f007:**
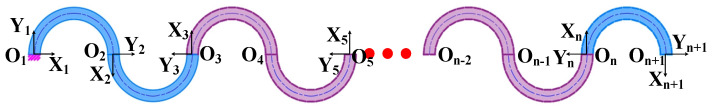
CF beam with odd number of links.

**Figure 8 micromachines-15-01039-f008:**
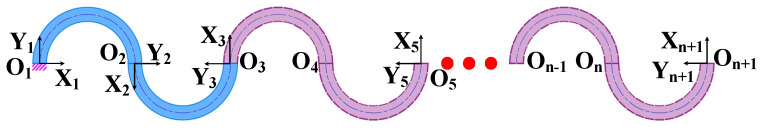
CF beam with even number of links.

**Figure 9 micromachines-15-01039-f009:**
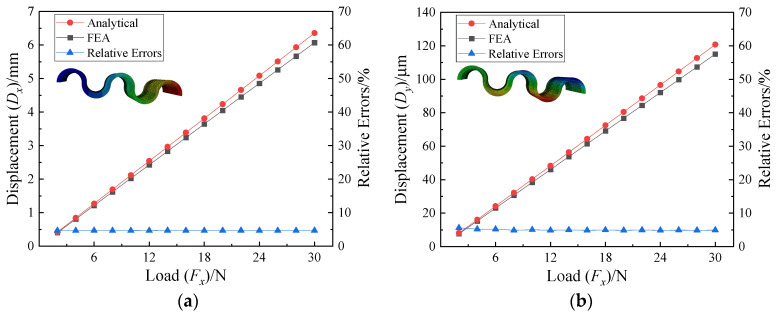
Displacement and relative error of free end of CF beam. (**a**) Displacement and relative error in x direction. (**b**) Displacement and relative error in y direction.

**Figure 10 micromachines-15-01039-f010:**
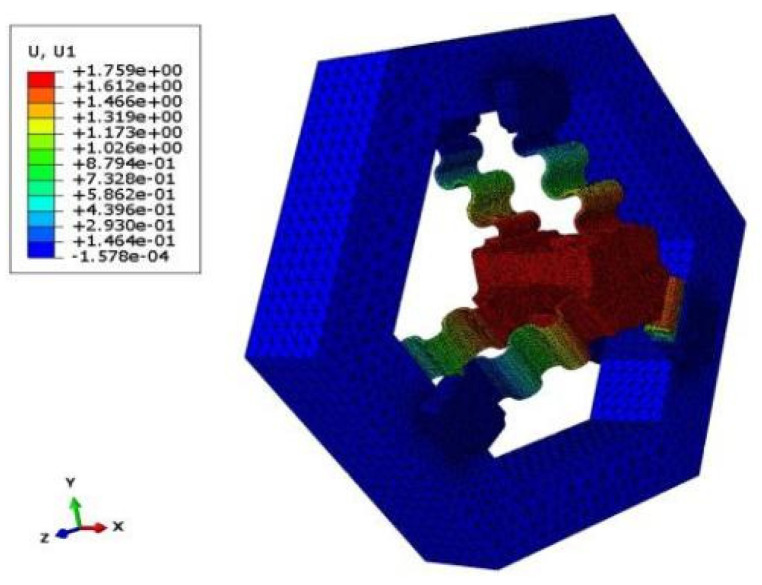
Displacement simulation of CF guide mechanism.

**Figure 11 micromachines-15-01039-f011:**
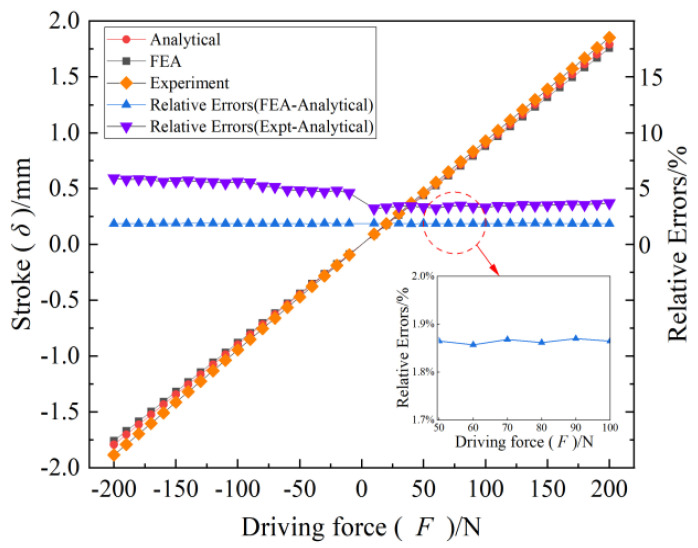
Displacement of FTS system under driving force.

**Figure 12 micromachines-15-01039-f012:**
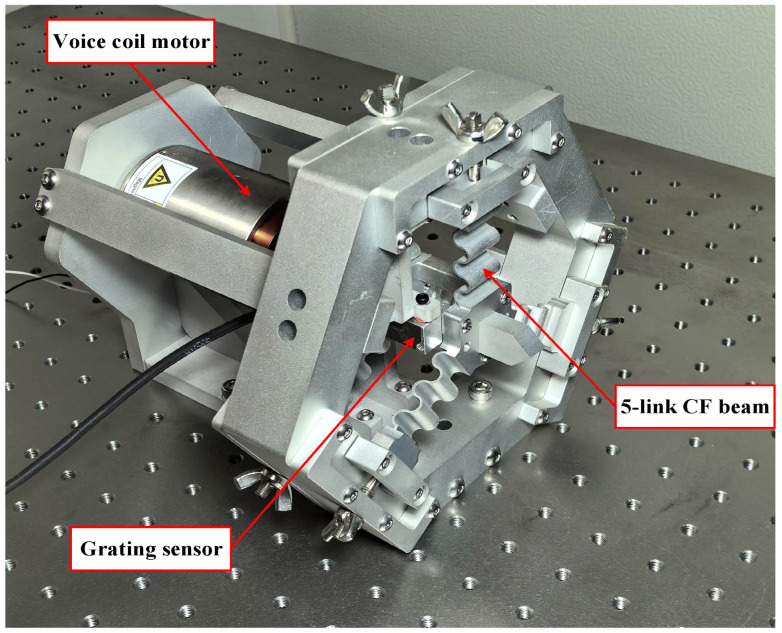
Five-link CF guiding mechanism FTS system.

**Figure 13 micromachines-15-01039-f013:**
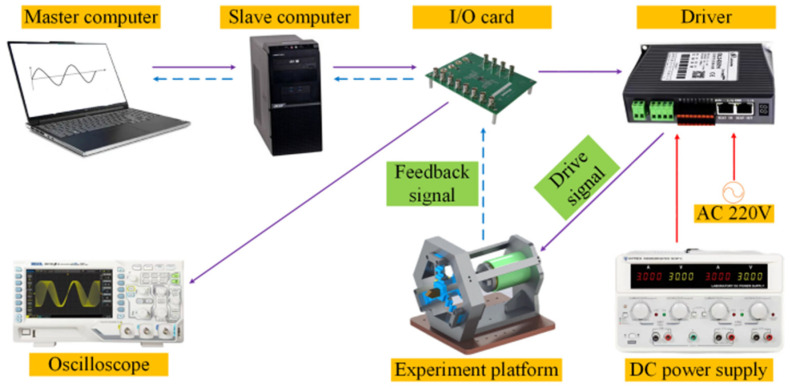
Experimental flow chart of five-link FTS system.

**Figure 14 micromachines-15-01039-f014:**
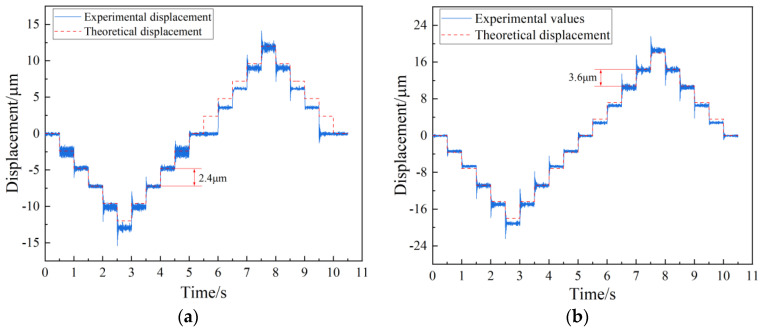
Positive and negative stroke resolution test. (**a**) Negative stroke resolution. (**b**) Positive stroke resolution.

**Figure 15 micromachines-15-01039-f015:**
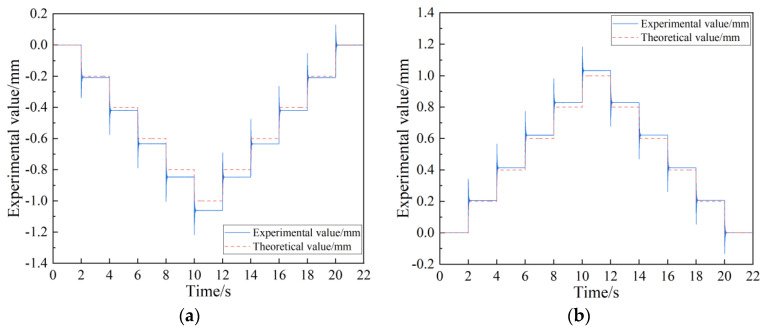
Positive and negative stroke positioning accuracy and repeat positioning accuracy test. (**a**) Negative stroke positioning and repetitive positioning tests. (**b**) Positive stroke positioning and repetitive positioning tests.

**Figure 16 micromachines-15-01039-f016:**
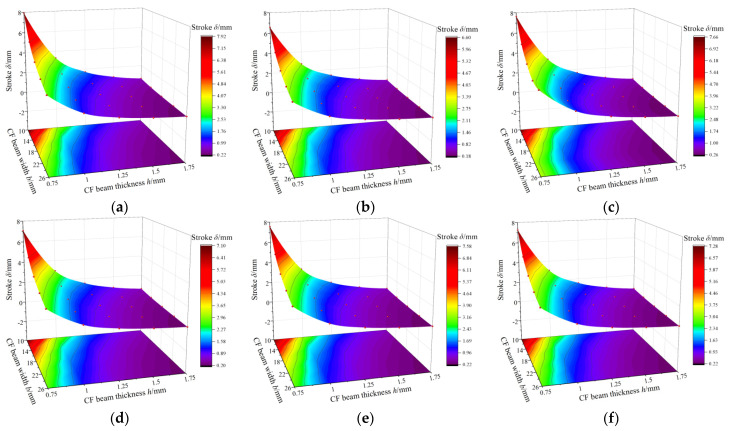
Structural design of CF guide plate in terms of stroke influence law. (**a**) Effect of *b* and *h* on stroke for *n* = 3. (**b**) Effect of *b* and *h* on stroke for *n* = 4. (**c**) Effect of *b* and *h* on stroke for *n* = 5. (**d**) Effect of *b* and *h* on stroke for *n* = 6. (**e**) Effect of *b* and *h* on stroke for *n* = 7. (**f**) Effect of *b* and *h* on stroke for *n* = 8.

**Table 1 micromachines-15-01039-t001:** Structural parameters of CF guiding mechanism.

Dimensional Parameters	Value (mm)
Inner radius of semi-circle flexure unit in CF beams (*r*_1_)	3
Outer radius of semi-circle flexure unit in CF beams (*r*_2_)	4
Width of CF beams (*b*)	18
Thickness of CF beams (*h*)	1
Length of CF beams (L)	35
Distance between two CF beams (Δ_x1_)	38
Distance between the end of the CF beam and the tool mounting surface (Δ_x2_)	4
Distance between the end of the CF beam and the axis of the guiding mechanism (Δ_y_)	17

**Table 2 micromachines-15-01039-t002:** CF guiding mechanism material parameters.

Density *ρ*/(kg/m^3^)	Young’s Modulus *E*/Mpa	Poisson’s Ratio *μ*	Tensile Strength/Mpa	Yield Strength/Mpa
2810	71,000	0.33	524	455

## Data Availability

The original contributions presented in this study are included in the article; further inquiries can be directed to the corresponding authors.

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
