# Peer review of "Design, Modeling, and Testing of a Long-Stroke Fast Tool Servo Based on Corrugated Flexure Units"

_micromachines, 2024, doi:10.3390/mi15081039_

Round 1
Reviewer 1 Report
Comments and Suggestions for Authors
Pleae, review:
A scientific article abstract is a concise and comprehensive summary of the research paper.
It typically includes the following components:
Background/Introduction: Briefly introduces the research topic, its significance, and the existing knowledge gap.
Objective: Clearly states the purpose of the study, the research question(s) being addressed.
Methods: Summarizes the research methodology, including study design, participants or subjects, data collection, and analysis techniques.
Results: Presents the main findings of the study in a concise and informative manner.
Conclusion: Offers a brief interpretation of the results, highlighting their implications and significance.
Intrpoduction
1 . Focus and Clarity: While the introduction covers a broad range of FTS technologies, it could benefit from a sharper focus on the specific problem the research addresses. Consider narrowing down the scope to emphasize the limitations of existing flexible mechanisms and how the proposed solution overcomes these challenges.
2. Research Gap:
The research gap could be more explicitly stated. A clear and concise sentence outlining the specific challenge that the proposed FTS system addresses would strengthen the introduction.
3. Contribution:
The introduction should clearly articulate the novel contribution of the research. While the proposed FTS system is mentioned, the specific advantages and innovations compared to existing solutions could be highlighted more explicitly.
4. Structure:
Consider reorganizing the introduction for better flow. For example, you could group the literature review based on different FTS technologies and their respective limitations. This would help to build a stronger case for the need for a new approach.
Author Response
We thank you for the positive comments. Please review the attachment.

Reviewer 2 Report
Comments and Suggestions for Authors
1. The authors declared that “Since the CF beam mainly undergoes bending deformation after being stressed, the effects of axial force and torque on the displacement of the free end can be neglected when applying the Mohr's” Therefore, the effects of axial force and torsion on the displacement of the free end can be removed from Eq. 1.
2. Figures in the text must appear in order. Figure 13 appears earlier than Figures 11 and 12.
3. The authors obtain some very important results from Figure 16. However, the authors did not explain in detail why?
4. This paper aims to design an innovative FTS with long stroke and high load-bearing capacity by utilizing the large deformation, high precision, and frictionless characteristics of CF beams. The flexibility matrix method and FEM were used to establish a static model of the 3-link FTS system and achieved good test results. And through this FTS system, the effectiveness of the test model of the 5-link FTS system experimental platform was confirmed. However, the authors did not state this clearly in the conclusion. So, conclusions should highlight the novelty and clearer advance in understanding presented in this study. Please try to put some quantifications, and comment on the limitations.
Author Response

(The authors gave the same response as above.)

Reviewer 3 Report
Comments and Suggestions for Authors
The paper present some interesting work on design, modelling and testing of a long-stroke fast tool servo based on corrugated flexure units. However, the paper manuscript needs to undertake the following revisions:
(1) The symbols in Nomenclature should be listed in the alphabetic order. The abbreviations should be also listed and included in Nomenclature.
(2) In section 5, the paper should provide a further clarification and discussion on the dynamic characteristics of the FTS, such as its bandwidth, natural frequency of the structure and mode analysis. They are essentially important for the FTS usage in ultraprecision and micro cutting applications.
(3) The following very relevant paper and book in the topic area should be reviewed and included in References section, particularly against the above comment (2):
- Smart cutting tools and smart machining: development approaches, and their implementation and application perspectives, Chinese Journal of Mechanical Engineering, 30(5), 2017, 1162-1176.
- Micro Cutting: Fundamentals and Applications, John Wiley & Sons, Chichester, October 2013.
Author Response

(The authors gave the same response as above.)

Round 2
Reviewer 2 Report
Comments and Suggestions for Authors
No.